# Microstructure Evolution of FeNiCoCrAl_1.3_Mo_0.5_ High Entropy Alloy during Powder Preparation, Laser Powder Bed Fusion, and Microplasma Spraying

**DOI:** 10.3390/ma14247870

**Published:** 2021-12-19

**Authors:** Anton Semikolenov, Pavel Kuznetsov, Tatyana Bobkova, Svetlana Shalnova, Olga Klimova-Korsmik, Viktor Klinkov, Ilya Kobykhno, Tatyana Larionova, Oleg Tolochko

**Affiliations:** 1Institute of Machinery, Materials and Transport, Peter the Great St. Petersburg Polytechnic University, 195251 St. Petersburg, Russia; semikolenov.a@edu.spbstu.ru (A.S.); o.klimova@ltc.ru (O.K.-K.); klinkov_va@spbstu.ru (V.K.); kobyhno_ia@spbstu.ru (I.K.); tolochko_ov@spbstu.ru (O.T.); 2Institute of Laser and Welding Technologies, State Marine Technical University, 190121 St. Petersburg, Russia; s.shalnova@corp.smtu.ru; 3The Federal State Unitary Enterprise “Central Research Institute of Structural Materials “Prometey”, Named by I.V. Gorynin of National Research Center “Kurchatov Institute”, 191015 St. Petersburg, Russia; prometey_35otdel@mail.ru (P.K.); bobkova_ti@crism.ru (T.B.)

**Keywords:** high entropy alloy, gas atomization, plasma spraying, laser powder bed fusion, thermal stability, microstructure

## Abstract

In the present study, powder of FeCoCrNiMo_0.5_Al_1.3_ HEA was manufactured by gas atomization process, and then used for laser powder bed fusion (L-PBF) and microplasma spraying (MPS) technologies. The processes of phase composition and microstructure transformation during above mentioned processes and subsequent heat treatment were analyzed by X-ray diffraction (XRD), scanning electron microscopy (SEM), energy dispersive X-ray spectroscopy (EDS), and differential thermal analysis (DTA) methods. It was found that gas atomization leads to a formation of dendrites of body centered cubic (BCC) supersaturated solid solution with insignificant Mo-rich segregations on the peripheries of the dendrites. Annealing leads to an increase of element segregations till to decomposition of the BCC solid solution and formation of σ-phase and B2 phase. Microstructure and phase composition of L-PBF sample are very similar to those of the powder. The MPS coating has a little fraction of face centered cubic (FCC) phase because of Al oxidation during spraying and formation of regions depleted in Al, in which FCC structure becomes more stable. Maximum hardness (950 HV) is achieved in the powder and L-PBF samples after annealing at 600 °C. Elastic modulus of the L-PBF sample, determined by nanoindentation, is 165 GPa, that is 12% lower than that of the cast alloy (186 GPa).

## 1. Introduction

Multicomponent alloys with high entropy began being studied in 2004 [1,2] and, until now, the amount of work devoted to high entropy alloys (HEA) continues to grow [3,4]. Even though the idea of HEA implied one-phase multicomponent solid solution alloys when it first arose, nowadays the concept of HEA includes a wide spectrum of multi-phase multicomponent alloys [5]. It was shown that multiphase HEA usually has higher strength compared to that of single-phase HEA [6,7,8,9]. Therefore, in recent works, attention has been paid to the structure evolution and the precipitation behavior of HEA [10,11,12,13,14,15,16,17]. The problem of structure and phase stability becomes especially acute when it concerns the alloys produced in non-equilibrium conditions. One of the most promising applications of HEA is powder production for additive technology [18,19,20] and spray coating [21,22,23,24]. For most powder technologies, raw powders should have spherical shape and narrow size distribution with an average size of 20–80 μm (depending on the method used). From this point of view, gas atomization can be considered the most appropriate powder-manufacturing route [25]. Gas atomization as well as additive and thermal spray techniques imply super-fast cooling of molten metal, which results in a highly non-equilibrium structure. In addition, the powder assembling processes includes the heating of an already applied layer, which may cause uncontrollable phase transformations and instability in material’s properties. Thus, structure evolution in HEA, especially those produced by superfast cooling of molten metal, is of great interest and importance.

The FeCoCrNiMoAl system was chosen for investigation due to its potentially high properties necessary for coating: high hardness, corrosion and wear resistance [26,27,28,29,30]. In cast condition, FeNiCoCrMo_0.5_Al consists of multicomponent B2 and σ phases. An increase of Ni over x = 1 [27] or a decrease of Al lower than x = 1 [26] leads to a formation of face centered cubic (FCC) solid solution and softening of the alloy. Cr and Mo induce σ-phase formation accounting for high hardness [28], but an increase of Cr higher x = 1 decreases the alloy’s fracture toughness [29]. An increase of Fe decreases wear resistance because of increased markedly oxidation rate [28]. In [26] it was shown that the FeNiCoCrMo_0.5_Al system has high potential for high-temperature structural application due to high hot hardness up to 1000 °C and softening resistance. The excellent softening resistance is explained as the result of the low diffusion rate of vacancies, which are trapped by different solute atoms in the multicomponent phases (B2 and σ) [26,27]. Early studies of FeNiCoCrMoAl HEA were focused on the investigations of cast alloys; studies on their microstructure under superfast cooling have not been yet performed [26,27,28,29,30]. In our previous work [30], it was found that the composition of FeCoCrNiMo_0.5_Al_1.3_ has the highest hardness and, therefore, was chosen for further investigation.

Thus, in this paper we analyze the processes of phase composition and microstructure transformation of FeCoCrNiMo_0.5_Al_1.3_ HEA during powder manufacturing and two methods of powder consolidation: laser powder bed fusion (LPBF) and micro-plasma spraying (MPS). The effect of annealing at 600 °C and 900 °C on the structure and microhardness of the manufactured samples was evaluated.

## 2. Materials and Methods

FeNiCoCrAl_1.3_Mo_0.5_ ingots of mass of approximately 50 g were melted in a high-frequency induction furnace in a quartz crucible under an argon atmosphere. Elemental components with a purity of 99.93% or greater were used. The melting process took roughly 2 min, and then the melt was cooled in water. Melting of refractory elements occurred via their gradual dissolution in the melt. For homogenization, the ingots were molten three times. Then the samples were annealed at 900 °C for 5 h. The prepared ingots were used for powder preparation by gas atomization process. The powder was produced using HERMIGA 75/3VI atomizer (Phoenix Scientific Industries Ltd., Hailsham, UK). About 3 kg of the alloy (60 ingots) was molten at 1600 °C for 10 min in induction-heated crucible, then the molten metal was poured through atomizer nozzle by argon gas with pressure of 40 bar [31]. Size distribution of the produced powder was analyzed by laser diffraction method using Malvern Mastersizer 2000 unit (Malvern Panalytical, Malvern, UK). Additive manufacturing of the powder was fulfilled by laser powder bad fusion (L-PBF). Cylindrical samples of 10 mm in diameter and 15 mm in height was manufactured using EOSINT M270 (L-PBF Solutions Group AG, Lübeck, Germany) in nitrogen atmosphere with laser power of 190 W and a scanning speed of 1000 mm/s, the spot size was 100 μm, and the melted layer thickness—40 μm. Coating was produced by microplasma spraying (MPS) of the powder to a stainless substrate (DIN 1.4843) using UGNP-7/2250 unit. The spraying was fulfilled at voltage of 38 V, argon plasma gas flow rate of 2.3 L/min (2300 slpm) and carrying argon gas flow rate of 2 L/min (2000 slpm). MPS is carried out using low power (up to 3 kW) plasmatron generating a quasi-laminar plasma jet at a current of 26 A. Upon MPS, the diameter of the sprayed material spot is 1.0 to 5.0 mm, which allows one to apply coatings on small parts and thin wall parts with no hazard of their shrinking and overheating. Spraying parameters were chosen empirically on the basis of preliminary experiments [32].

Heat treatment of the samples was carried out at a temperature of 600 °C and 900 °C for 1 h in air atmosphere. The phase composition was examined with the D8 Advance diffractometer (Bruker, Billerica, MA, USA) using Cu *K*_α_ radiation. The solid solution lattice parameter was determined on the base of XRD patterns, which had been collected in the 2Θ range from 20° to 140° with a speed of 2°/min. For microscopic observation and the hardness test, the samples were embedded in a resin and polished. Scanning electron microscopy (SEM) and energy dispersion spectroscopy (EDS) were performed on a MIRA 3 (TESCAN, Brno, Czech Republic) microscope with AztecLive.Advanced.Ultim.Max.65 EDS detector (Oxford instruments, Abingdon, UK). DTA curves were obtained at heating at 20°/min up-to 900 °C in air. Vickers microhardness was tested with a FM-310 (FUTURE-TECH CORP., Japan) under a load of 50 g and a dwelling time of 10 s, at least seven measurements per point were made. The equilibrium phases were calculated in the Thermo-Calc program using the database “TCHEA 4: High Entropy Alloy v 4.0”. The accuracy and calculation speed settings correspond to the standard values. Nanoindentation was performed with Hysitron TI 750 Ubi Nanoindenter (Bruker, Billerica, MA, USA) using Berkovich indenter under a load of 1500 μN, with loading time of 10 s and unloading time of 10 s. Mapping and distribution of hardness and elastic modulus throughout the surface of area of 900 μm^2^ were obtained by indentation matrix of 8 × 8 indents.

## 3. Results and Discussion

SEM images of the powder particles are presented in Figure 1a,b. The particles have spherical shape and narrow particle’s size distribution typical for gas atomized powders [33]. Mean particle’s diameter is about 25 μm (Figure 1c). Figure 1d presents EDS spectrum of the powder. Chemical compositions of the powder and the average compositions of the ingots used for powder preparation are presented in Table 1. As it is seen the chemical composition of the prepared powder matches that of the cast alloy and close to the nominal one.

Microstructure of the cast alloy after homogenization, equilibrium phase composition calculated in the Thermo-Calc, and XRD patterns of the cast and atomized samples are presented in Figure 2. Microstructure of the cast alloy contains primary dendrites of BCC_B2, enriched with Al and Ni, and complex interdendritic structure consisted of BCC_B2 and σ-phase. Inside of the dendrites one can see thin laminar precipitates of σ-phase [30] (Figure 2a,c). According to Thermo-Calc the alloy in the equilibrium state at ambient condition should consist of BCC_B2, σ, and μ-phase, (Figure 2b), however, μ-phase was not found. The absence of the μ-phase may be caused by both extremely low formation rate due to low diffusive mobility at its equilibrium temperature interval or insufficient reliability of the database. XRD pattern of the gas atomized powder contains only BCC reflexes, (Figure 2c), however, each BCC peak consists of several ones, which are partly overlapped (Figure 2d). It is obviously caused by the coexisting of several BCC solid solutions with different lattice parameter, due to element segregation [27,28,29,30]. XRD pattern of the bulk sample fabricated by L-PBF is similar to that of the powder. XRD pattern of the coating apart from BCC peaks contains also minor FCC reflexes. The phases, their chemical compositions, determined by EDS, and lattice parameters, determined by XRD, are presented in Table 2.

Microstructures of the powder, the sample fabricated by L-PBF, and the coating deposited by MPS are presented in Figure 3. The microstructure of the powder is dendritic, slight contrast indicates insignificant liquation. Microstructure of the L-PBF sample is similar to that of the powder, but with more pronounced liquation of heavy element (Mo) on the peripheries of the dendrites. The observations are in an agreement with XRD results denoting presence of several BCC solid solutions slightly varying by the compositions and, correspondingly, by the lattice parameters (Figure 2d).

The coating has a more complex and inhomogeneous structure: one can see deformed particles distinguished from each other. It may be noted, that depending on the size, the particles have different phase contrast: the smallest particles are lighter due to lower Al content. It was suggested in [22,23] that during plasma spraying, Al interacts with oxygen, which leads to the formation of both aluminum oxide layers on the particle surface and regions depleted in Al. At Al concentration in FeCoNiCrAlMo_0.5_ lower than 10% a formation of FCC solid solution become preferable [26,30], this fact explains presence of FCC reflexes in the XRD pattern. The large particles have liquation similar to that observed in the powder microstructure. The middle size particles have no phase contrast, and their chemical composition, determined by EDS, corresponds to the average composition of the alloy; it is possible that their cooling rate was high enough to prevent solution decomposition.

Figure 4a shows DTA curve for the powder. Three wide exothermic peaks can be distinguished in the temperature intervals of 600–750 °C and 800–900 °C. In general, the appearance of exothermic peaks during heating of a non-equilibrium structure indicates a transition to a more stable state (with lower Gibbs energy) due to overcoming the energy barrier. In our case, this means decomposition of the supersaturated solid solution and σ-phase precipitation. According to Thermo-Calc (Figure 2b), there are no equilibrium phase transformations throughout the interval of 250–1300 °C and even the molar fractions of the phase remain almost the same, corresponding to 60% of BCC_B2 and 40% of σ-phase, which is confirmed by absence of the endothermic effects on DTA curve. XRD patterns of the powder and L-PBF sample annealed at 900 °C for 1 h are presented in Figure 4b. Both materials consist of BCC_B2 and σ-phase, similar to the phase composition of the cast and homogenized alloy (Figure 2c). Lattice parameters of the BCC_B2 in the annealed powder and L-PBF sample are 2.881 ± 0.001 and 2.887 ± 0.003 Å, correspondingly, which coincides with to that of the cast alloy (2.886 ± 0.005 Å).

Microstructures of the materials annealed at 600 °C and 900 °C are presented in Figure 3. Element distributions in the powder sample annealed at 600 °C and 900 °C are presented in Figure 5. There are no noticeable changes in the microstructures after annealing at 600 °C compared to the initial state. After annealing at 900 °C, decomposition of the solid solution becomes obvious. The σ-phase precipitates can be divided into two types: the first are formed in the areas that was enriched with Mo, they have bigger size (up to 500 nm) and form continuous (in the powder) or discontinuous (in the L-PBF sample) net. The second type—fine (up to 100 nm) roundish precipitates homogeneously distributed inside the areas enriched with Al and Ni.

Figure 6 demonstrates the results of the microhardness test. In initial state, the powder has higher microhardness than cast alloy (760 HV). Despite the fact that no microstructural changes were observed after annealing at 600 °C (Figure 3), microhardness of the powder and L-PBF sample annealed at 600 °C grows by about 25% compared to that in the initial state. This hardness rise may be caused by a growth of elastic stresses between the several BCC solid solutions due to increasing element segregation. After annealing at 900 °C microhardness slightly decreases but remains higher than that in the initial state. From one hand, the decomposition of the solid solution results in stress relaxation and hardness decrease, from other the precipitation of hard and homogeneously distributed σ-phase holds the hardness on a high level (about 850 HV). The powder and L-PBF sample annealed at 900 °C have similar hardness despite the difference in precipitates morphology (Figure 3). The high temperature heat treatment of the MPS coating led to a partially destruction of the sample. The phenomenon is often observed in coating particularly those deposited by plasma spraying and linked to a formation of “diffusion cell”—aluminum-depleted particles diffusionaly isolated from the bulk [34]. In a consequence, the protective capacity of the alumina layer is impaired, and voluminous oxides of Cr, Fe, Ni, Mo, Co fast grown, which leads to stress development and failure.

Nanoindentation was performed to determine the mechanical properties of the separate phases and analyze their distribution. Typical load-depth curves and the distribution of hardness and modulus are shown in Figure 7.

L-PBF fabricated sample demonstrates much more narrow distribution of the properties due to single-phase structure, some value dispersion is due to dendritic liquation. High dispersion of hardness and modulus of the cast alloy (Figure 7b) reflects its complex two-phase microstructure. However, predominantly bimodal distributions of the properties (Figure 7c,d) allows to evaluate the properties of each phase. It should be noted that the elastic modules of the phases in the cast alloy are higher than that of the solid solution in the L-PBF fabricated sample. BCC phases in L-PBF sample are predominantly disordered solid solutions (Table 2) with metallic bonding; in contrast to them, the cast alloy is composed of BCC_B2 and σ-phase, both are intermetallic compounds with partially covalent rigid bonding, which imparted higher value of elastic modulus to the cast alloy.

## 4. Conclusions

In the present study, the powder of FeCoCrNiMo_0.5_Al_1.3_ HEA was synthesized via gas atomization and then used for laser powder bed fusion and microplasma spraying processes. Microstructure and phase composition stability were studied.

FeCoCrNiMo_0.5_Al_1.3_ in equilibrium state consists of B2 and σ-phase. Gas atomization led to a formation of partially ordered BCC super saturated solid solution with insignificant Mo-rich and Al-rich segregations. The alloy remained single-phase after powder consolidation via laser powder bed fusion procedure. The non-equilibrium solid solution decomposes with B2 and σ-phase formation only after annealing at 900 °C for 1 h.

Maximal microhardness (950 HV) was obtained for the powder and the L-PBF sample annealed at 600 °C, which is caused by increasing element segregation and growth of elastic stresses. After annealing at 900 °C microhardness slightly decreased (850 HV) but remained higher than that in the initial state, it is explained by from one hand stress relaxation, from other homogeneous precipitation of hard σ-phase. 

The MPS coating had a more inhomogeneous structure. During spraying, Al interacts with oxygen, which led to the formation of both oxide layers on the particle surface and regions depleted in Al, in which BCC structure becomes unstable and FCC solid solution forms. High temperature annealing at 900 °C led to properties deterioration.

The elastic modulus of the L-PBF sample, determined by nanoindentation, is 165 GPa, that is 12% lower than that of the cast alloy (186 GPa).

## Figures and Tables

**Figure 1 materials-14-07870-f001:**
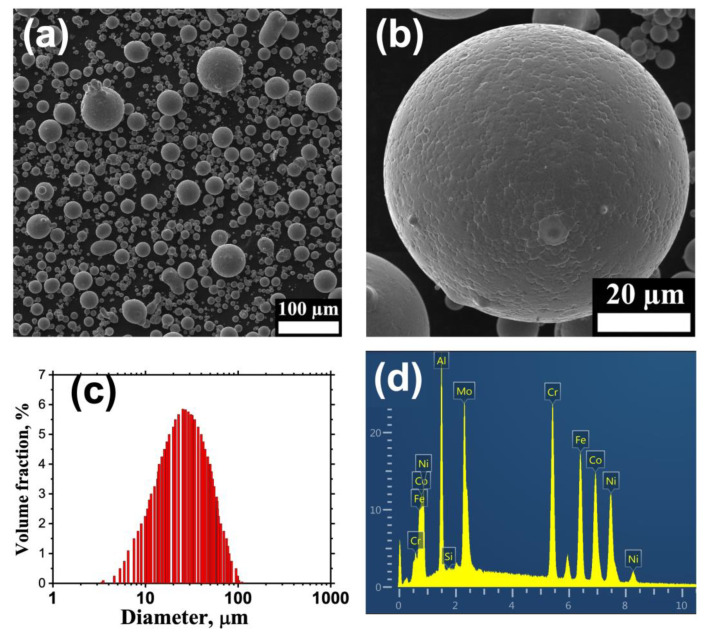
SEM image of gas atomized FeNiCoCrAl_1.3_Mo_0.5_ powder (**a**,**b**); particle size distribution (**c**); EDS spectrum of the powder (**d**).

**Figure 2 materials-14-07870-f002:**
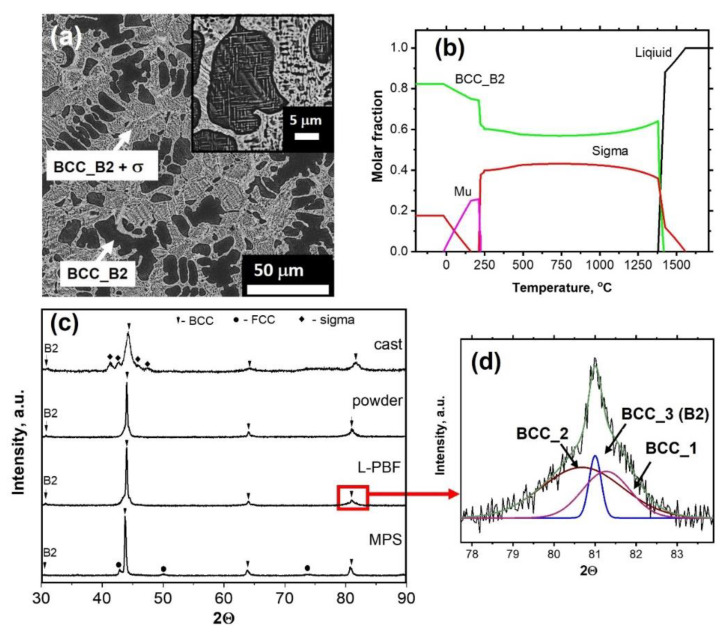
Microstructure of the cast alloy (**a**), equilibrium phases in dependence on the temperature calculated by Thermo-Calc (**b**), XRD patterns of the cast alloy, powder, L-PBF, and MPS fabricated samples (**c**), demonstration of XRD peak deconvolution (**d**).

**Figure 3 materials-14-07870-f003:**
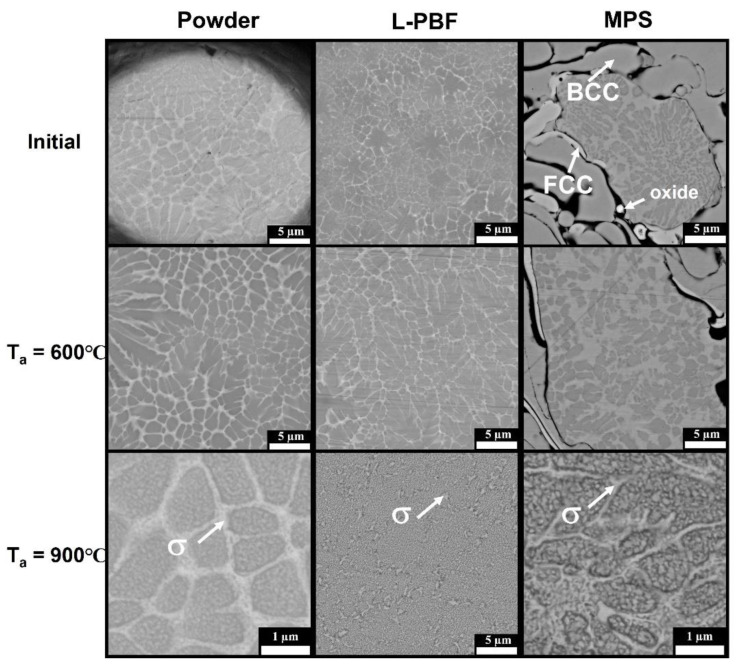
SEM microphotographs of the powder, L-PBF and MPS samples in an initial state and annealed at 600 °C and 900 °C.

**Figure 4 materials-14-07870-f004:**
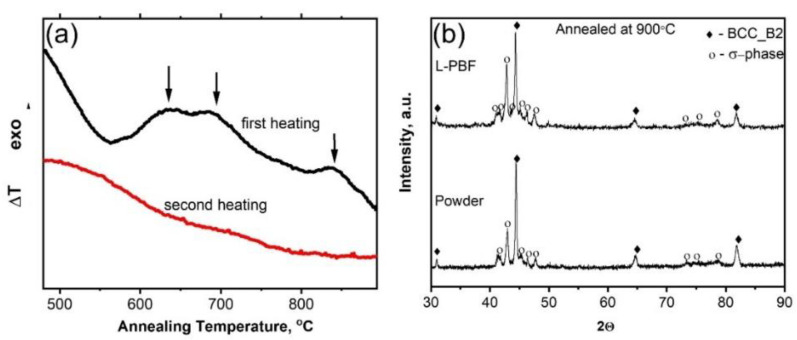
DTA curves of the powder (**a**) and XRD patterns of the powder and L-PBF sample after annealing at 900 °C for 1 h (**b**).

**Figure 5 materials-14-07870-f005:**
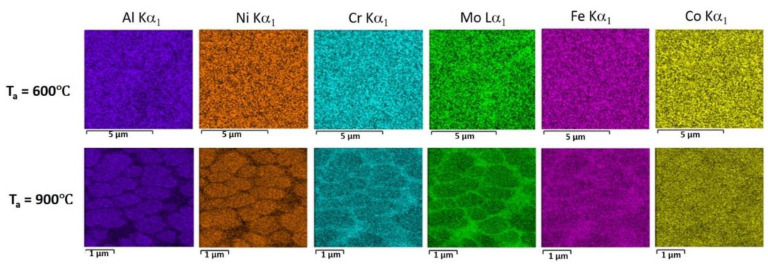
EDS mapping of the powders annealed at 600 °C and 900 °C.

**Figure 6 materials-14-07870-f006:**
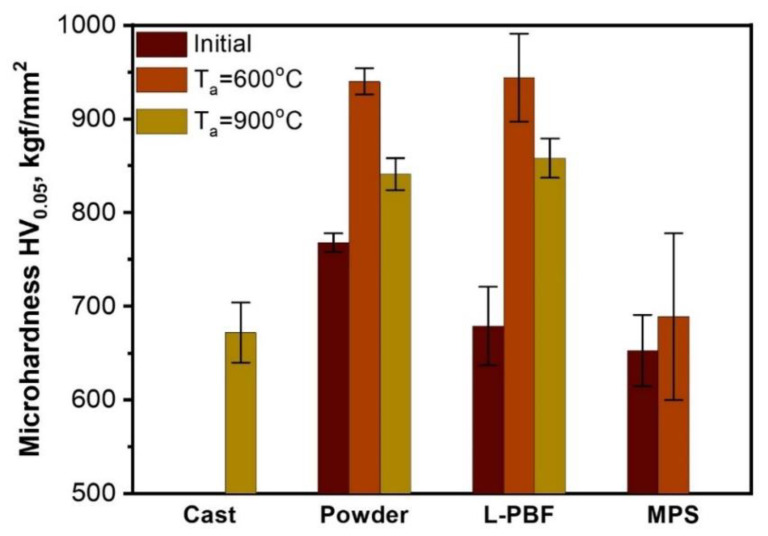
Vickers microhardness of the samples in an initial and annealed state.

**Figure 7 materials-14-07870-f007:**
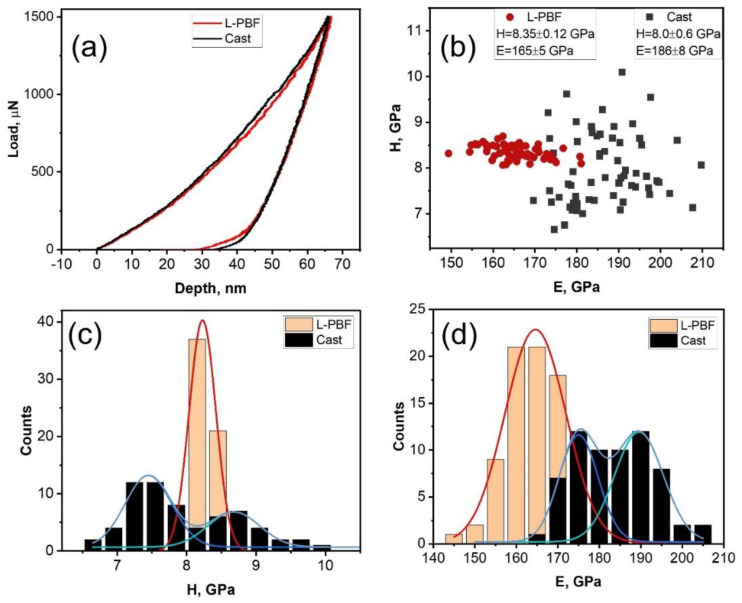
Typical load-depth curves (**a**), hardness vs. modulus plot for each indentation (**b**), hardness (H) (**c**) and modulus (E) (**d**) distributions for the cast and L-PBF samples.

**Table 1 materials-14-07870-t001:** Chemical composition of the cast sample and the gas atomized powder.

Sample	Atomic Concentration, %						
	Fe	Ni	Co	Cr	Mo	Al	Si
Cast	17.2	17.3	17.1	17.0	9.3	22.1	0.14
Powder	17.2	17.3	17.2	16.9	9.3	22.0	0.19

**Table 2 materials-14-07870-t002:** Phase composition.

Sample	Phase	Atomic Concentration, * %						Lattice Parameter, Å
		Fe	Ni	Co	Cr	Mo	Al	
Cast and homogenized	BCC_B2 Al,Ni-rich	12.5	**24.3**	**18.2**	10.4	3.4	**31.2**	2.886 ± 0.005
	Interdendrite σ + BCC_B2	**21.2**	11.5	16.8	**25.2**	**11.1**	14.3	-
Powder	BCC_1	17.7	17.1	17.4	17.5	9.3	21.0	2.889 ± 0.005
	BCC_2 Mo-rich	17.5	16.9	17.4	**17.9**	**10.0**	20.3	2.939 ± 0.006
	BCC_3 (B2) Al-rich	17.2	16.6	16.9	17.1	9.4	**22.9**	2.909 ± 0.001
MPS coating	BCC_1	17.8	17.1	17.5	17.4	9.1	21.1	2.878 ± 0.008
	BCC_2 Mo-rich	**17.7**	16.1	16.7	**18.5**	**12.0**	18.6	2.932 ± 0.005
	BCC_3 (B2) Al-rich	16.9	16.9	17.0	16.9	9.8	**22.6**	2.907 ± 0.003
	FCC	**20.7**	**20.8**	**21.1**	17.6	11.7	8.0	3.632 ± 0.002
L-PBF	BCC_1	17.9	17.1	17.5	17.5	9.0	21.0	2.878 ± 0.008
	BCC_2 Mo-rich	17.5	15.6	17.2	**18.0**	**12.8**	18.9	2.932 ± 0.005
	BCC_3 (B2) Al-rich	17	16.3	16.6	16.8	9.7	**23.7**	2.909 ± 0.002

*—bold figures correspond to the enhanced (over average) values.

## Data Availability

Not applicable.

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
