# Peer review of "Microstructure Evolution of FeNiCoCrAl1.3Mo0.5 High Entropy Alloy during Powder Preparation, Laser Powder Bed Fusion, and Microplasma Spraying"

_materials, 2021, doi:10.3390/ma14247870_

Round 1
Reviewer 1 Report
The present work focuses on the microstructure evolution of FeNiCoCrAl1.3Mo0.5 High Entropy Alloy during powder manufacturing and two methods of powder consolidation: laser powder bed fusion and microplasma spraying. Being multi-component and also multi-phase the problem of the microstructure and phase stability of these alloys becomes especially important when they are produced in non-equilibrium conditions.
Additionally the effect of annealing at two different temperatures on the structure and microhardness and elastic moduli of the alloys was evaluated as well.
In my opinion this is a very comprehensive microstructural study on important for the practice new materials and the results obtained would be of interest for the researchers in the field of metal science. The manuscript can be published in the present form with some improvements:
- Since the initial alloys were prepared by heating in a quartz crucible it is possible to contain Si. It case this is true it is better to be indicated in table 1.
- In the conclusions it is not necessary to mention the methods for structural and microstructural characterization. This information is enough to be presented in the Abstract.
- The summary as a whole could be shortened – the most important results to be presented.
Author Response
Thank you for the review of our work and positive report!
Below are our response for your comments:
- Since the initial alloys were prepared by heating in a quartz crucible it is possible to contain Si. It case this is true it is better to be indicated in table 1.”
Si concentration was added in the Table 1.
- In the conclusions it is not necessary to mention the methods for structural and microstructural characterization. This information is enough to be presented in the Abstract.
- The summary as a whole could be shortened – the most important results to be presented.
We rewrote the conclusion taking into account your comment.
Reviewer 2 Report
Larionova et al studied microstructure evolution of FeNiCoCrAl1.3Mo0.5 high entropy alloy during powder preparation, laser powder bed fusion, and microplasma spraying. The work presents interesting results and can be considered for acceptance after minor revisions.
1. In the introduction part, the author should briefly introduce the research of other research groups on FeCoNiCrMoAl high entropy alloys.
2. Figure 3. Particle sizes in different materials should be described.
3. TEM images with different magnifications are recommended to provide to better reflect the characteristics of the material. But this one is optional.
4. Figure 5. Spaces should be left between numbers and units. It is "1 μm" rather than "1μm".
5. There are some formatting problems in the references. Check whether letters should be uppercase or lowercase, and whether numbers should be subscripts.
Author Response
Thank you for revision of our work!
Below are our response to your comments:
- In the introduction part, the author should briefly introduce the research of other research groups on FeCoNiCrMoAl high entropy alloys.
The introduction has been improved, information about this system presented by other researches was added.
- Figure 3. Particle sizes in different materials should be described.
For all samples the same powder was used with mean particle’s diameter of about 25 μm, the powder particle size distribution was presented in Fig.1,c. On the microphotographs of the powder in initial state a particle of about 30 μm is shown, in annealed at 600℃ a particle of about 35-40 μm is shown, and the same for the powder annealed at 900 ℃. According to our observation, the particles have similar structure independent on the size.
- TEM images with different magnifications are recommended to provide to better reflect the characteristics of the material. But this one is optional.
We agree that TEM observation would be interesting and useful, unfortunately, right now we have no options to use TEM. Nowadays we perform research on aging of these alloys and will try to use TEM for characterization.
- Figure 5. Spaces should be left between numbers and units. It is "1 μm" rather than "1μm".
It was corrected
- There are some formatting problems in the references. Check whether letters should be uppercase or lowercase, and whether numbers should be subscripts.
It was corrected
Reviewer 3 Report
In the present work the Authors declared investigation of microstructure evolution of FeNiCoClAl1.3Mo0.5 HEA's during powder preparation, laser powder bed fusion and microplasma spraying. Investigation was performed in very sytematic way and used analytical techniques seems to be adequte for the work. However, during reading, a few questions and comments arrises as follows:
- Abstract: In this section a number of shortcuts was used, e.g. XRD, SEM, EDS, BCC, L-BPF etc. Since this section is placed as one of the first, these abbreviation should be explained. Please consieder.
- Page 5, lines 152-163: The Authors described the microstructure of MPS coating and noticed, that it consists of "separated powder particles". Please comment in more detail what is the reason of such microstructure? It seems that the powder particles were not sufficiently molten during spraying, or maybe the speed was not optimal? These are guestations, and this should be known from your text. Please provide more detailed description of MPS process. Please add also comment what could be the possible consequences of such microstructure during coating service?
- Page 7, line 206: The Authors describes the microhardness of materials and mentioned, that heat treatment in air at 900C "led to degradation of the properties". The question is why? The Authors mentioned that heat-treatment was performed in air atmosphere. Since in the case of MPS coating a separated powder particles were observed, thus it is highly possible that during air exposure at 900C so-called "diffusion cell effect" has occured (e.g. H.E. Evans, M.P. Taylor, Diffusion cells and chemical failure of MCrAlY bond coats in thermal-barrier coating systems, Oxid. Met. 55 (2001) 17–34, https://doi.org/10.1023/A:1010369024142.). Then the question is, why heat-treatment was done in air, not under vacuum or at least in an innert atmoshere, to avoid oxidation phenomenon?
- Page 8, Figure 6: It is clear that the hardness for powder and coatings (L-PBF and MPS) increases after heat-treatment at 600C, but for powder and L-PBF decrease after HT at 900C. hat is the reason responsible for this? Please comment.
- Page 9, chapter 4: In the present form this is rather summary of observations than Conclusions. Plea re-write this chapter accordingly.
General comments:
- I do not see clearly the motivation for performing the present work, i.e. clear reason why this work has been done. Maybe if the Authors could propose potential application od produced coatings (L-PBF and MPS) it could help? Please consider.
- Comment about MPS coatings: as mentioned the obtained microstructure of MPS coatings is not promissing in term of its potential application. Is there any way to improve the MPS process to obtain coatings with better microstructure? Please comment.
- The title of present wok is: "Microstructure Evolution of FeNiCoCrAl1.3Mo0.5 High Entropy Alloy During Powder Preparation, laser powder bed fusion, and microplasma spraying". Thus, I understand that the evolution of HEA microstructure during powder preparation, L-PBF and MPS shall be investigated. However, the point is, that mentioned microstructure should evolve from something. the most logical way is to show evolution of HEA microstructure from "cast" rod of HEA used for powder preparation and further processes. Menawhile, microstructure of cast ros is missing, thus there is no starting poing of microstructure evolution. Finally, the article title is missleading. Therefore, please add the microstructure of cast rod, or change the title accordingly.
Despite my comments I found this work worth publishing and I recommend to reconsider its publication after major revision.
Round 2
Reviewer 3 Report
The Authors seriously reacted on Reviewers' comments and improved the article quality. Thus I think that it can be published in the present version.